# Newly Designed Poxviral Promoters to Improve Immunogenicity and Efficacy of MVA-NP Candidate Vaccines against Lethal Influenza Virus Infection in Mice

**DOI:** 10.3390/pathogens12070867

**Published:** 2023-06-23

**Authors:** Martin C. Langenmayer, Anna-Theresa Luelf-Averhoff, Lisa Marr, Sylvia Jany, Astrid Freudenstein, Silvia Adam-Neumair, Alina Tscherne, Robert Fux, Juan J. Rojas, Andreas Blutke, Gerd Sutter, Asisa Volz

**Affiliations:** 1Institute for Infectious Diseases and Zoonoses, LMU Munich, 80539 Munich, Germany; martin.langenmayer@gmx.net (M.C.L.); anna.luelf@googlemail.com (A.-T.L.-A.); lisa.marr@klinikum-nuernberg.de (L.M.); sylvia.jany@viro.vetmed.uni-muenchen.de (S.J.); astrid.freudenstein@viro.vetmed.uni-muenchen.de (A.F.); s.adamneumair@campus.lmu.de (S.A.-N.); alina.tscherne@viro.vetmed.uni-muenchen.de (A.T.); robert.fux@lmu.de (R.F.); jrojas@ub.edu (J.J.R.); gerd.sutter@lmu.de (G.S.); 2German Center for Infection Research (DZIF), Partner Site Munich, 80539 Munich, Germany; 3Institute of Clinical Hygiene, Medical Microbiology and Infectiology, Paracelsus Medical University, Klinikum Nürnberg, 90419 Nuremberg, Germany; 4Immunology Unit, Department of Pathology and Experimental Therapies, Faculty of Medicine and Health Sciences, University of Barcelona—Bellvitge Biomedical Research Institute (IDIBELL), 08908 Barcelona, Spain; 5Research Unit Analytical Pathology, Helmholtz Zentrum Munich, 85764 Neuherberg, Germany; andreas.parzefall@lmu.de; 6Institute for Veterinary Pathology, LMU Munich, 80539 Munich, Germany; 7Institute of Virology, University of Veterinary Medicine Hannover, 30559 Hannover, Germany; 8German Center of Infection Research (DZIF), Partner Site Hannover-Braunschweig, 30559 Hannover, Germany

**Keywords:** influenza A virus, nucleoprotein, Modified Vaccinia virus Ankara, vaccine, synthetic VACV promoter, CD8+ T cell response

## Abstract

Influenza, a respiratory disease mainly caused by influenza A and B, viruses of the *Orthomyxoviridae*, is still a burden on our society’s health and economic system. Influenza A viruses (IAV) circulate in mammalian and avian populations, causing seasonal outbreaks with high numbers of cases. Due to the high variability in seasonal IAV triggered by antigenic drift, annual vaccination is necessary, highlighting the need for a more broadly protective vaccine against IAV. The safety tested Modified Vaccinia virus Ankara (MVA) is licensed as a third-generation vaccine against smallpox and serves as a potent vector system for the development of new candidate vaccines against different pathogens. Here, we generated and characterized recombinant MVA candidate vaccines that deliver the highly conserved internal nucleoprotein (NP) of IAV under the transcriptional control of five newly designed chimeric poxviral promoters to further increase the immunogenic properties of the recombinant viruses (MVA-NP). Infections of avian cell cultures with the recombinant MVA-NPs demonstrated efficient synthesis of the IAV-NP which was expressed under the control of the five new promoters. Prime-boost or single shot immunizations in C57BL/6 mice readily induced circulating serum antibodies’ binding to recombinant IAV-NP and the robust activation of IAV-NP-specific CD8+ T cell responses. Moreover, the MVA-NP candidate vaccines protected C57BL/6 mice against lethal respiratory infection with mouse-adapted IAV (A/Puerto Rico/8/1934/H1N1). Thus, further studies are warranted to evaluate the immunogenicity and efficacy of these recombinant MVA-NP vaccines in other IAV challenge models in more detail.

## 1. Introduction

Influenza virus epidemics constitute a worldwide major public health threat. Annual outbreaks of influenza A virus (IAV) and influenza B virus are responsible for considerable morbidities in the human population. Especially elderly people and infants are at increased risk for a severe disease course and increased mortality [1,2]. The annual recurrence of these viruses is mainly caused by processes known as antigenic drift and antigenic shift [3]. Antigenic drift refers to the viruses’ ability to accumulate mutations in the genes encoding two major surface proteins, hemagglutinin (HA) and neuraminidase (NA). These mutations enable immune escape by altered recognition of the virus-neutralizing antibodies. Antigenic shift refers to the recombination of HA and NA gene subtypes, resulting in the generation of a new reassortant virus. Besides human IAV, reassortment can also include subtypes of avian or swine IAV with increased pandemic potential, because the human population can be immunologically naïve regarding these new viruses [4,5].

The currently used inactivated seasonal influenza vaccines mainly induce antiviral antibodies against the HA protein to neutralize virus attachment and entry [6]. Due to the high variability of this antigen, the usefulness of these vaccines is limited in the case of the circulation of a different influenza virus strain than was predicted [7]. Influenza vaccines are currently updated annually to provide protection given the drifting antigens of the viruses. Vaccine failure results from strain mismatches and reassortant potentially pandemic viruses. Thus, more universal influenza vaccines which provide protection against or a reduction in severe disease from heterologous influenza strains are needed [8,9].

In this approach, the nucleoprotein (NP), as the conserved internal protein, has gained interest because of its moderate variation between IAV strains. Some strains have more than 90% amino acid overlap of the NP between different HA-subtypes [10,11,12]. Moreover, the NP contains potent dominant epitopes, which are targeted by CD4+ and CD8+ T-cells [13,14], and is thus considered an important T-cellular antigen. Especially the virus-specific CD8+ T cells are important for heterologous protection between different IAV strains [15].

In addition, mucosal antibodies against IAV are able to neutralize virions early and are probably capable of blocking infection completely and, thus, preventing transmission [16]. This does not refer to anti-NP antibodies, because the NP is only expressed after infection. However, anti-NP antibodies might protect against severe disease by preventing the spread of virions within the body.

One approach towards a more broadly protective universal influenza vaccine is to elicit strong T cell immunity in combination with robust antibody responses to improve the reactivity and longevity of the IAV-specific immune response.

Modified Vaccinia virus Ankara (MVA), a highly attenuated and replication-deficient poxvirus, has been used as a viral vector to generate vaccines against various infectious diseases such as MERS [17], COVID-19 [18,19], or Ebola [20]. Although evaluation of these vaccines in clinical trials has revealed an excellent safety and immunogenicity profile with the induction of both cellular and humoral immune responses [21,22,23,24,25], research on the improvement of MVA immunogenicity is still ongoing (for review: [26]). One approach focuses on the modification of vaccinia virus (VACV) promoters that drive the expression of the recombinant target antigen, aiming to improve the antigen-specific immune responses [27,28]. 

VACV promoters are classified into early, intermediate, and late elements, depending on the cascade-like timing of the gene expression throughout the poxviral infection [29]. VACV promoters [30,31] which show early and late elements allow for the expression of viral genes during the early and late phase of viral replication. Several natural or synthetic promoters for the expression of foreign antigens driven by the poxviral transcription machinery were described previously, including the natural early PVGF, late P11 promoters, and the modified early/late promoter PmH5 [32,33]. As early poxviral proteins are described to elicit strong CD8+ T cell responses [34,35,36], it is hypothesized that early VACV promoters are also linked to strong T cell responses towards heterologous antigens [36,37,38].

In this context, we aimed to design new synthetic promoters for recombinant gene expression combining early and late viral promoter elements and to evaluate these chimeric VACV-specific promoters in recombinant MVA candidate vaccines producing an IAV NP antigen. Using single shot and prime-boost vaccinations, we demonstrate the induction of strong immune responses and the protective capacity of MVA vector vaccines using different promoter systems. Different readouts show protective immunity during acute IAV infection and allow us to correlate vaccine-induced protection with CD8+ T cell responses.

## 2. Materials and Methods

### 2.1. Cell Cultures

Primary chicken embryonic fibroblasts (CEF) were prepared from 10- to 11-day-old chicken embryos (SPF eggs, VALO, Cuxhaven, Germany) and were maintained in Minimum Essential Medium Eagle (MEM) (Sigma-Aldrich, Taufkirchen, Germany) containing 10% heat-inactivated fetal bovine serum (FBS) (Sigma-Aldrich, Taufkirchen, Germany) and 1% MEM non-essential amino acid solution (Sigma-Aldrich, Taufkirchen, Germany). Madin-Darby Canine kidney (MDCK) cells (ATCC CCL-34) were cultured in MEM (Sigma-Aldrich, Taufkirchen, Germany) containing 10% heat-inactivated FBS and 1% Penicillin–Streptomycin (Pen/Strep) (Sigma-Aldrich, Taufkirchen, Germany). Cells were maintained at 37 °C and 5% CO_2_ atmosphere.

### 2.2. Plasmid Construction

The coding sequence of the full-length influenza A virus (IAV) nucleoprotein (NP) (A/Puerto Rico/8/1934/H1N1); GenID: 956531) was modified in silico to remove guanine or cytosine runs and termination sequences for VACV-specific early transcription. Furthermore, cleavage sites for the restriction endonucleases HpaI and NotI were added to the sequence. The modified cDNA sequence was generated by DNA synthesis (GeneWiz, Leipzig, Germany) and was cloned into the MVA transfer plasmid pIIIred, under transcriptional control of either the strong natural early promoter PVGF or the synthetic chimeric promoters PLMU1 (PII + PVGF), PLMU2 (PI + PmH5), PLMU3 (P11 + PVGF), and PLMU4 (P11 + PmH5) (pIIIred-PLMU1-NP, pIIIred-PLMU2-NP, pIIIred-PLMU3-NP, pIIIred-PLMU4-NP, pIIIred-PVGF-NP).

### 2.3. Generation of Recombinant MVA Vector Viruses

Recombinant MVA delivering NP under control of either the synthetic or natural promoters (PLMU1-4 and PVGF, respectively) were generated as described previously [19]. In brief, MVA (clonal isolate MVA-F6-sfMR) served as a backbone virus to construct the recombinant MVA vector viruses. CEF cells at 80–90% confluence were infected with MVA at a multiplicity of infection (MOI) of 0.05 and were transfected with the above-described vector plasmids using X-treme Gene PD DNA Transfection Reagent (Roche Diagnostics, Penzberg, Germany) according to the manufacturer’s instructions. Cells were collected 48 h post-infection and recombinant viruses were clonally isolated by serial plaque passages using the co-expression of the fluorescent protein marker mCherry. To obtain vaccine preparations, recombinant MVA vector viruses were amplified on CEF cell monolayers, purified by ultracentrifugation through 36% sucrose cushions, and reconstituted in 10 mM Tris-HCl buffer (pH 9.0) to high-titer stock preparations. Viral titers were determined by counting plaque-forming units (pfu).

Quality control of recombinant MVA vector viruses was performed in compliance with standardized in-house protocols [19,39]. PCR analysis of genomic viral DNA was used to confirm genetic stability and identity of the newly generated viruses. Multi-step growth experiments using permissive (CEF) and non-permissive cells (human HaCat) were conducted to test replicative capacity of the recombinant viruses. 

### 2.4. Western Blot Analysis of Recombinant Proteins

CEF cells were infected with recombinant MVA viruses at a MOI of 1. Cell lysates were prepared at 0, 4, 8, 12, 24, and 48 h post-infection and were stored at −80 °C. Samples were resolved by sodium dodecyl sulfate (SDS)-polyacrylamide (10%) gel electrophoresis (SDS-PAGE), and proteins were transferred onto nitrocellulose membranes by wet electroblotting. Membranes were blocked with PBS/T (0.1% Tween20) containing 5% non-fat dried milk powder (Carl Roth, Karlsruhe, Germany) for 1 h at room temperature. Afterwards, membranes were probed overnight at 4 °C with a primary antibody (monoclonal mouse anti-NP (1:100), Biozol, Eching, Germany, or ß-Actin (1:2000), Taufkirchen, Germany) diluted in blocking buffer. Membranes were washed with PBS/T and were probed with goat-anti-mouse IgG conjugated to horseradish peroxidase (HRP) (1:5000; Agilent Dako, Glostrup, Denmark) for 1 h at room temperature. After washing the membranes with PBS/T, SuperSignal^®^ West Dura Extended Duration substrate (Thermo Fisher Scientific, Planegg, Gemany) was added for development. Blots were visualized using MicroChemi 4.2 imager (DNR Bio-Imaging Systems, Neve Yaming, Israel). 

### 2.5. Mouse Immunization and Infection Experiments

Specific pathogen-free 6–10-week-old C57BL/6 mice (in-house breed) were housed in isolated cage units (Techniplast, Hohenpeißenberg, Germany) with access to food and water ad libitum. The experiments were approved by the Government of Upper Bavaria, Munich, Germany, and were carried out in accordance with the German regulations for animal experimentation (Animal Welfare Act). 

Groups of mice were immunized twice over a 21-day interval with recombinant MVA vaccines at a dose of 10^8^ pfu intramuscularly in the hind legs. Mice immunized with non-recombinant MVA (MVA) at a dose of 10^8^ pfu and saline (PBS) served as controls. Twenty-eight days after booster vaccination, mice were challenged with a lethal dose (10^3^ TCID_50_) of a mouse-adapted influenza A virus (A/Puerto Rico/8/1934/H1N1) by nasal inoculation. Upon infection, mice were monitored for clinical signs and weighed, and experiment was terminated 28 days after challenge infection or after humane endpoints were reached.

Groups of mice were immunized once with recombinant MVA vaccines at a dose of 10^8^ pfu intramuscularly in the hind legs. Mice immunized with non-recombinant MVA (MVA) and saline (PBS) respectively served as controls. Twenty-eight days after prime vaccination, mice were challenged with a lethal dose (10^3^ TCID_50_) of a mouse-adapted influenza A virus (A/Puerto Rico/8/1934/H1N1) by nasal inoculation. A dose of 10^3^ TCID_50_ was used, since this infection reproducibly resulted in fatal disease in >90% of the mice. After challenge infection, mice were monitored daily at least twice for well-being, health constitution, and clinical signs such as habitus of fur, posture, anorexia, lethargy/depression, and respiratory symptoms using a clinical score sheet. Weights of all mice were checked daily. The experiment was terminated eight days or four weeks post-challenge infection or after humane endpoints were reached. 

### 2.6. Depletion of Specific Subsets of Immune Cells

Depletion of CD8+ T cells was performed as described previously [40]. In brief, 100 µg of a monoclonal mouse anti-CD8 antibody (clone 2.43, Harlan Bioproducts, Indianapolis, USA) was injected via the intraperitoneal route on days −2 and −1 prior to challenge infection, four weeks after prime immunization. Successful depletion of CD8+ T cells was tested by flow-cytometric analysis of blood cells from antibody-treated mice.

### 2.7. Necropsy, Histology and Immunohistochemistry

After euthanasia, mice were subsequently necropsied. The lung and a comprehensive organ panel were macroscopically examined. The percentage of lung affected by pneumonia was macroscopically scored using a five-step scoring system (not affected [0%]; affected by 25%, 50%, 75%, and 100%). After determining lung weight, the lung lobes were dissected for histology and viral titration. The accessory lobe was deep frozen at −80 °C for viral titration (see Section 2.9). The other organs were processed for histology following published guidelines [41,42,43]. After paraformaldehyde-fixation, paraffin-embedding, and cutting, sections were stained with hemalum-eosin. For NP detection via immunohistochemistry, a modified mouse-on-mouse protocol [44] after proteinase K incubation was used. Primary mouse monoclonal anti-NP antibodies (clone HB65) were incubated in vitro with a biotinylated goat anti-mouse IgG Fab fragment (JacksonImmunoResearch Laboratories) and subsequently saturated with mouse normal serum. This cocktail was added to the slides and the reaction was developed as previously described [45]. Positive (murine IAV-infected lung) and negative (first antibody replaced with irrelevant one) controls were routinely performed to confirm specificity of staining.

### 2.8. Quantitative Stereologic Examinations

The volumes of inflamed lung tissue compartments within the left lung lobe were analyzed in two healthy, two mock-vaccinated, and six vaccinated mice using unbiased quantitative stereological analysis methods. Left lobe was fixed by bronchial infusion with 4% neutrally buffered formaldehyde solution at a transpulmonary pressure of 20 cm H_2_O. The bronchus was ligated when the flow ceased [46]. Thereafter, lobes were placed in fresh fixative for 12 h while intrapulmonary pressure was still maintained. Afterwards, lung volume was estimated using a submersion method [47]. After paraffin-embedding and microtome calibration, the lung was exhaustively sectioned in equidistant sections. Per lobe, 10–12 sections were sampled using a systematic uniform random (SUR) sample scheme. The relative volumes of the different lung compartments within the lung were determined from the fractional areas of their section profiles and the area of total pulmonal tissue in HE stained histological sections. The section areas of inflamed and non-inflamed tissue compartments were determined by point counting in up to 21 SUR-selected fields of view (FOV) per section at 200x [48]. In total, 2277 ± 546 points were counted per case. The total volumes of different tissue compartments were calculated from their respective volume fractions within the lung lobe and the total lung lobe volume.

### 2.9. Determination of IAV Loads in Mouse Lung Lobes

Left lung lobes obtained at necropsy were weighed and homogenized with a tissue lyser (Retsch Tissue Lyser MM300, Quiagen GmbH, Hilden, Germany). Subsequently, lungs were centrifuged twice for 1 min with 1500 rpm at 4 °C and supernatant was transferred into a fresh tube. Tenfold serial dilutions were prepared and 90% confluent MDCK cells were inoculated with the virus samples for 1 h at 33–37 °C. Afterwards, cells were washed twice with plaque assay wash medium (Dulbecco’s Modified Eagle’s Medium (DMEM) Sigma-Aldrich, Taufkirchen, Germany, +1% Pen/Strep). Then, 2-fold plaque assay medium (DMEM, 3% Pen/Strep, 4 mM L-glutamine (Thermo Fisher Scientific, Planegg, Germany), 50 mM HEPES (Sigma-Aldrich, Taufkirchen, Germany), 2 µg/mL TPCK-treated trypsin (Sigma-Aldrich, Taufkirchen, Germany)) was mixed in a 1:1 ratio with 1.6% low melting point agarose (Biozym Scientific GmbH, Oldendorf, Germany) and was added as an overlay to the cells. After solidification of the overlay, plates were incubated for 72 h at 33–37 °C. The agar overlay was removed, and cells were stained with crystal violet. Pulmonal viral titers were determined by counting plaque-forming units (pfu).

### 2.10. Quantification of NP-Specifc IgG Antibodies by Enzyme-Linked Immunosorbant Assay (ELISA)

For analysis of IAV-NP specific serum IgG titers, flat-bottom 96-well ELISA plates (Nunc. MaxiSorp. Plates, Thermo Scientific) were coated with 50 ng/well recombinant NP protein [49] and were incubated over night at 4 °C. Plates were blocked with PBS containing 1% bovine serum albumin (BSA) (Sigma-Aldrich, Taufkirchen, Germany) and 0.15 M sucrose (Sigma-Aldrich, Taufkirchen, Germany) for 1 h at 37 °C. Mice sera were three-fold serially diluted in PBS containing 1% BSA (PBS/BSA), starting at a 1:30 dilution. NP pre-coated ELISA-plates were incubated with the diluted mice sera for 1 h at 37 °C. Subsequently, plates were washed with PBS/T (0.05% Tween20), probed with goat anti-mouse IgG HRP (1:2000, Agilent Dako, Denmark) diluted in PBS/BSA, and developed with 3,3′,5,5′-tetramethylbenzidine (TMB) (Sigma-Aldrich, Taufkirchen, Germany). The absorbance was measured at 450 nm with a 620 nm reference wavelength. Total antibody titers were calculated as described previously [50].

### 2.11. CD8+ T-Cell Analysis by Enzyme-Linked Immunospot Assay (ELISpot)

Spleens were harvested at necropsy and splenocytes were prepared as described previously [51]. In brief, spleens were passed through a 70 µm strainer (Falcon^®^, A Corning Brand, Corning NY, USA) and incubated with Red Blood Cell Lysis Buffer (Sigma-Aldrich, Taufkirchen, Germany). Splenocytes were resuspended in RPMI 1640 medium (Sigma-Aldrich, Taufkirchen, Germany) containing 10% heat-inactivated FBS and 1% Pen/Strep and were stimulated with the NP-specific peptide ASNENMETM [NP_366–374_] [52,53]. Non-stimulated cells and cells stimulated with phorbol myristate acetate/ionomycin (PMA) or the vaccinia virus specific peptide TSYKFESV [B8_20–27_] [35] served as controls. IFN-γ-producing cells were measured by IFN-γ ELISpot assay using the IFN-γ ELISpotPLUS kit (Mabtech, Stockholm, Sweden) according to the manufacturer’s protocol. Automated ELISPOT plate reader software (A.EL.VIS Eli. Scan, A.EL.VIS ELISPOT Analysis Software, Hannover, Germany) was used to count and analyze the spots.

### 2.12. T Cell Analysis in Blood Using FACS Analysis

Mice were bled on day 0, 7, 28, 36, and 56 after initial vaccination. A total of 50 µL of heparinized blood was preincubated with 10 µL of NP-specific dextramer (Immudex, Copenhagen, DK) for 15 min at room temperature. After preincubation, anti-mouse CD3 phycoerithrin (PE)-Cy7 (clone 17A2, 1:100, Biolegend), anti-mouse CD4 phycoerithrin (PE)-Cy7 (clone GK1.5, 1:600, Biolegend), and anti-mouse CD8α Alexa Fluor 488 (clone 53–6.8, 1:300, Biolegend), using 50 µL/sample diluted in FACS buffer, was added. After 30 min on ice, blood samples were incubated with Red Blood Cell Lysis Buffer (Sigma-Aldrich, Taufkirchen, Germany). Cells were washed and resuspended in FACS Buffer. Data were acquired by the MACSQuant VYB Flow Analyser (Miltenyi Biotec) and analyzed using FlowJo (FlowJo LLC, BD Life Sciences, Ashland, OR, USA).

### 2.13. Data Analysis

Statistical tests and calculations were performed with Microsoft Excel (Microsoft Office 2019, Redmond, WA, USA) and GraphPad Prism version 5 (GraphPad Software Inc., San Diego, CA, USA). Data were analyzed by Mann–Whitney test unless otherwise indicated. A *p* < 0.05 was regarded to be statistically significant.

## 3. Results

### 3.1. Design and In Vitro Testing of Recombinant MVA Vaccines Delivering NP by Chimeric Promoters 

We designed four synthetic vaccinia virus early/late promoters to test their capacity to activate robust NP-specific cellular and humoral responses compared to the strong early natural PVGF promoter. NP gene sequences were introduced into the deletion site III of MVA by homologous recombination, and placed under the control of the newly designed chimeric promoters PLMU1, PLMU2, PLMU3, and PLMU4, as well as the natural PVGF promoter, as already established (Figure 1a,b) [38]. The genetic integrity of the recombinant viruses was confirmed by the PCR analysis of the viral DNA, demonstrating the site-specific insertion of the IAV-NP gene sequence, the absence of non-recombinant MVA, and the proper removal of the mCherry marker gene (Appendix A). Furthermore, the genetic stability of the recombinant MVA viruses was confirmed by PCR targeting the six major deletion sites and the C7L gene locus of MVA (Appendix A). 

The recombinant viruses replicated efficiently in chicken embryo fibroblasts (CEF), but not in the human HaCat cells (Appendix A). To further characterize the expression pattern of the recombinant IAV-NP, the total cell lysates from CEF cells infected with the recombinant MVA viruses were analyzed by Western Blot. The mouse monoclonal antibody directed against the NP revealed one prominent protein band that migrated with molecular masses of ~56 kDa (Figure 1c). In general, NP synthesis was detectable at 8 h post-infection (hpi) and increased over 24 to 48 hpi. Of note, in cell lysates from MVA-PVGF-NP-infected cells, a first NP band was visible at 4 hpi (Figure 1c).

### 3.2. Immunogenicity and Protective Efficacy of MVA-NPcandidate Vaccine after a Prime-Boost Vaccination Regimen

To evaluate the immunogenicity and efficacy of the MVA candidate vaccines, we designed vaccination-challenge studies which compared different immunization strategies. In the first experiment, C57BL/6 mice were vaccinated twice, 21 days apart, with non-recombinant MVA or saline (PBS) as controls or with one of the five MVA vaccines (10^8^ pfu). Four weeks after booster vaccination, the mice were challenged with a lethal dose of a mouse-adapted influenza A virus (A/Puerto Rico/8/1934/H1N1) by nasal inoculation. The experiment was terminated and all mice were necropsied at day 28 after the IAV challenge or after humane endpoints were reached. All of the mice from the two control groups showed clinical abnormalities (ruffled fur, hunched position, and body weight loss) and had to be euthanized at day 8 after the challenge infection (Figure 2a,b). The necropsy at day 8 after the challenge revealed multifocally reddened and consolidated lungs (suggestive of pneumonia) with increased lung scores (Figure 2c). Furthermore, in histologic pulmonic sections, we could confirm a pneumonia with epithelial necrosis, an influx of granulocytes and macrophages, alveolar edema, and septal necrosis (Figure 2d). Immunohistochemistry detected the NP antigen in intact and necrotic epithelia (Figure 2d), as well as in macrophages and alveolar septa.

However, the mice vaccinated with the recombinant MVA vaccines exhibited improved clinical outcomes after the lethal IAV challenge, except for two mice from the MVA-PVGF-NP-vaccinated group, which died at day 11 after the challenge due to clinical disease with severe respiratory symptoms and substantial weight loss. At necropsy, the two euthanized MVA-PVGF-NP-vaccinated mice exhibited severe pneumonia, with IAV detection in one animal. All other mice vaccinated with the MVA-PVGF-NP candidate vaccine were sufficiently protected from death; however, they developed different degrees of weight loss eight days after the challenge, with the lowest values ranging between 72% and80% of their initial body weight. Likewise, mice immunized with the MVA-PLMU-NP candidate vaccines continuously lost weight six–eight days after the challenge, with MVA-PLMU1-NP demonstrating the highest weight loss (lowest value: 82% of initial body weight) compared to MVA-PLMU2-NP, MVA-PLMU3-NP, MVA-PLMU4-NP (lowest body weight value: 89%, 84%, and 88% of initial body weight, respectively). 

At necropsy, the viral titration revealed infectious virus in 2/3 PBS and 3/3 non-recombinant MVA-vaccinated mice and one animal that was immunized with MVA-PVGF-NP. MVA-PLMU-NP-vaccinated animals did not display detectable infectious virus, which was in line with the mostly unremarkable gross pathology 28 days after the challenge, with only a few reddened pulmonic areas (Figure 2e). The serologic analyses at the end of the experiment revealed comparable levels of anti-NP serum IgG for MVA-PVGF-NP, MVA-PLMU1-NP, MVA-PLMU2-NP, and MVA-PLMU3-NP, with mean titers of 1:19,163, 1:20,850, 1:21,850, and 1:19,995, respectively. The mice immunized with MVA-PLMU4-NP showed lower serum IgG antibodies, with a mean titer of 1:6131 (Figure 2f).

To assess the activation of IAV-specific cellular immunity, we monitored NP-specific CD8+ T cells in the immunized and IAV-infected mice. Splenocytes were prepared upon necropsy and the cells were re-stimulated with the influenza A virus (A/Puerto Rico/8/1934/H1N1) H2-D^b^-restricted peptide NP_366–374_ [52,53]. Immunization with the recombinant MVA vaccines induced NP_366–374_-specific CD8+ T cell responses with mean numbers of 127 IFN-γ spot-forming cells (SFCs) in 10^6^ splenocytes for MVA-PVGF-NP, 92 SFCs for MVA-PLMU1-NP, 95 SFCs for MVA-PLMU2-NP, 66 SFCs for MVA-PLMU3-NP, and 78SFCs for MVA-PLMU4-NP (Figure 2g).

### 3.3. Immunogenicity and Protective Efficacy of MVA-NP Candidate Vaccines Using a Prime Vaccination Regimen

Since our recombinant MVA vaccines were confirmed to be protective in the IAV-mouse model using a prime-boost immunization model, we tested whether the protection against a lethal IAV challenge is sufficient after a single vaccination. Therefore, mice were vaccinated once with the recombinant MVA candidate vaccines and were infected with a lethal dose of IAV (A/Puerto Rico/8/1934/H1N1) 28 days post-vaccination (d.p.v.). Again, the mice were sacrificed 28 days after challenge infection or after humane endpoints were reached. The mice immunized with non-recombinant MVA or saline displayed similar signs of disease as observed before and had to be euthanized eight days after the challenge. The mice vaccinated with the recombinant MVA vaccines were protected against a lethal IAV challenge. The mice body weight curves displayed a similar course when compared to the previous experiment (Figure 3a,b). The weight loss of the MVA-PVGF-NP-vaccinated mice was, again, more pronounced than in the other groups. The lung scores did not show statistically significant differences between the recombinant MVA-vaccinated groups (Figure 3c). The lung histology of these animals displayed no acute lesions and was comparable to the first experiment. The immunohistochemistry did not reveal the NP antigen in any of the vaccinated mice and, furthermore, no infectious virus was detectable in the lungs from the vaccinated animals (Figure 3d). The serological analyses at the end of the experiment revealed comparable serum IgG titers for MVA-PVGF-NP, MVA-PLMU2-NP, MVA-PLMU3-NP, and MVA-PLMU4-NP, with mean titers of 1:17,987, 1:16,560, 1:17,712, and 1:12,613, respectively. The mice immunized with MVA-PLMU1-NP showed lower serum IgG antibodies, with a mean titer of 1:9095. (Figure 3e). However, this was not statistically significant when compared to the other recombinant MVA groups. 

The induction of cellular immune responses after prime immunization and the IAV challenge were tested as described above. Splenocytes were prepared upon necropsy and restimulated with the CD8+ T cell epitope NP_366–374_ [52,53]. Immunization with MVA-PVGF-NP and MVA-PLMU4-NP induced higher CD8+ T cell responses (mean 64 SFCs and 43 SFCs, respectively) compared to immunization with MVA-PLMU1-NP, MVA-PLMU2-NP, and MVA-PLMU3-NP (mean 37 SFCs, 32 SFCs, and 19 SFCs, respectively (Figure 3f).

Since all four chimeric promoters induced a protective effect against the lethal IAV challenge and produced comparable antibody and T-cell responses in the first vaccination experiments, we aimed to analyze the activation of peripheral blood NP-specific CD8+ T cell expansion in a next step. Mice were prime vaccinated and challenged 28 d.p.v. The levels of NP-specific CD8+ T cells were analyzed on days 0, 7, 28, 36, and 56 after their initial immunization. The control vaccinated animals did not mount NP-specific CD8+ T cells. In contrast, we confirmed T cell expansion in all MVA-NP-vaccinated animals. In MVA-PVGF-NP-, MVA-PLMU1-NP-, MVA-PLMU2-NP-, and MVA-PLMU3-NP-vaccinated mice, we detected similar levels of NP-specific CD8+ T cells starting on day 36 after the initial vaccination. On day 36, we detected a mean of 1525 NP-specific CD8+ T cells in MVA-PVGF-NP-vaccinated animals, a mean of 1297 NP-specific CD8+ T cells in MVA-PLMU1-NP-, 2326 NP-specific CD8+ T cells in MVA-PLMU2-NP-, and a mean of 2178 NP-specific CD8+ T cells in MVA-PLMU3-NP-vaccinated animals (Figure 4) [54]. These levels remained constant until day 56. Interestingly, in MVA-PLMU4-NP-vaccinated animals, we had already detected NP-specific CD8+ T cells seven days after the initial vaccination, with a mean of 1306 NP-specific CD8+ T cells. These titers further increased until day 28 after the initial vaccination, with a mean of 4448 NP-specific CD8+ T cells. On day 36 post-initial vaccination, the MVA-PLMU4-NP-vaccinated animals mounted a mean of 5867 NP-specific T cells, which then finally maintained with a mean of 8343 NP-specific CD8+ T cells on day 56. These data indicated that MVA-PLMU4-NP induced considerably higher levels of NP-specific CD8+ T cells in the blood. 

### 3.4. Immunogenicity Is Already Established at d8 p.i. in Selected Vaccines after Prime Vaccination

Next, based on the results from the immunogenicity and efficacy testing, we aimed to further characterize the rapidly protective capacity of the selected promising MVA-NP candidate vaccines in more detail early after lethal IAV challenge infection. For this, we again performed single vaccination with MVA-PLMU4-NP and challenged all mice 28 d.p.v. For comparison, MVA-PVGF-NP expressing the NP under the strong PVGF early promoter was included as a candidate vaccine. The morbidity and mortality were monitored, as in previous experiments. Eight days after the challenge, all mice were euthanized and analyzed for their viral load, immune responses, and pathology (Figure 5). All mice from the two control groups displayed weight loss, and one mouse from the MVA control group had to be euthanized seven days after the challenge. (Figure 5a). The mice immunized with MVA-PVGF-NP displayed weight loss without any clinical symptoms but recovered and started to gain weight. The mice immunized with MVA-PLMU4-NP displayed initial weight loss but recovered and were able to regain their initial body weight eight days after the challenge infection. The relative lung weights were significantly lower in the MVA-PVGF-NP- and MVA-PLMU4-NP-immunized mice compared to the PBS- and MVA-vaccinated mice (Figure 5b). In addition, the relative lung weights of the MVA-PLMU4-vaccinated mice were significantly lower than those of the MVA-PVGF-NP-vaccinated mice (Figure 5b). Likewise, the lung scores were significantly lower in recombinant MVA-NP-vaccinated mice compared to the control groups, and the lung scores of the MVA-PLMU4-vaccinated mice were significantly lower than those of the MVA-PVGF-NP-vaccinated mice (Figure 5c). To complement the subjective lung scoring with objective quantitative stereological estimation, the absolute volume of the inflamed left lung lobe was determined. The absolute volume of the inflamed left lung lobe was significantly reduced in the vaccinated mice compared to the control groups. The absolute volume of the inflamed lung parenchyma was significantly correlated with the relative left lung lobe weight (Spearman’s r = 0.75; *p* = 0.0098) (Figure 5d).

Lung viral loads were detected in 7/9 (77.8%) control mice, in 4/7 (57.1%) MVA-PVGF-NP-vaccinated mice, and in 1/9 (11.1%) MVA-PLMU4-NP-vaccinated mice (Figure 5e). The vaccinated mice displayed significantly reduced lung viral loads when compared to the control mice. The lung viral loads between the MVA-PVGF-NP- and MVA-PLMU4-NP-vaccinated mice showed no obvious or statistically significant differences. 

A serological analysis revealed similar levels of anti-NP serum IgG in MVA-PVGF-NP- and MVA-PLMU4-NP-immunized mice, with mean titers of 1:14,445 and 1:13,015, respectively (Figure 5f). Furthermore, the induction of CD8+ T cell responses upon re-stimulation of splenocytes with the NP-specific epitope NP_366–374_ could be confirmed in the MVA-PVGF-NP- and MVA-PLMU4-NP-immunized mice, with means of 43 SFCs and 66 SFCs, respectively. (Figure 5g).

### 3.5. Role of CD8+ T Cells for the Protective Capacity of MVA-PVGF-NP and MVA-PLMU4-NP Induced Protection 

To evaluate the role of CD8+ T cells in the outcome of protective immunity after a single vaccination with MVA-PVGF-NP or MVA-PLMU4-NP, the mice were depleted of CD8+ T cells before IAV challenge 28 d.p.v., as established before. The experiment was terminated 28 days after the challenge infection or after humane endpoints were reached. The mice vaccinated with MVA-PLMU4-NP or MVA-PVGF-NP were protected and survived, as expected from the previous experiments. MVA-PVGF-NP-vaccinated and CD8+ T cell-depleted mice succumbed to the IAV infection within a similar time frame to the control mice. The MVA-PLMU4-NP-vaccinated and CD8+ T cell-depleted mice displayed non-significant (area under the curve comparison) weight loss after the challenge but regained weight thereafter and survived until the experiment’s termination (Figure 6a). The lung weights and scores were significantly different between the MVA-PVGF-NP- and MVA-PLMU4-NP-vaccinated and CD8+ T cell-depleted mice (Figure 6b,c). The lungs of the MVA-PVGF-NP-vaccinated and CD8+ T cell-depleted mice displayed red pneumonic areas upon macroscopic examination (Figure 5d). In histologic sections, the lungs displayed epithelial necrosis, the influx of granulocytes and macrophages, alveolar oedema, and septal necrosis. Immunohistochemistry detected the NP antigen in intact and necrotic epithelia and macrophages in 4/5 animals. The necropsy findings of the other groups were comparable to previous experiments. Lung viral load could be detected in the control mice and MVA-PVGF-NP-vaccinated and CD8+ T cell-depleted mice (Figure 6e). Serological analysis revealed higher serum IgG titers in the MVA-PVGF-NP-vaccinated and CD8+ T cell-depleted mice, MVA-PVGF-NP-vaccinated mice, or MVA-PLMU4-NP-vaccinated mice, with mean titers of 1:26,723, 1:24,141, and 1:21,357, respectively. The MVA-PLMU4-NP-vaccinated and CD8+ T cell-depleted mice showed anti-NP serum IgG antibodies with a mean titer of 1:38,095 (Figure 6f). 

## 4. Discussion

In the current study, we aimed to evaluate the effect of newly composed, VACV-specific promoters on the immunogenicity and efficacy of MVA-based candidate vaccines. We used the influenza A virus nucleoprotein as a model antigen in a mouse model for lethal IAV challenge infection. NP represents a relatively conserved viral protein across IAV strains and, therefore, serves as a promising target to induce cross-protective cytotoxic T-cells against heterologous IAV infection [14,15,55]. Hereby, the protective immunity against IAV is considered to predominantly involve CD8+ T cells targeted to different NP epitopes [56]. More surprisingly, antibodies directed against NP also show antiviral activity and contribute to long-lasting anti-IAV antibody responses [57,58]. The hypothesis of this study was that the use of composed natural or modified promoters would result in an improved delivery and presentation of NP antigen which, subsequently, would strengthen the activation of IAV-specific immune responses. 

Using this approach, we confirmed that robust NP-specific immune responses after vaccination with a recombinant MVA-NP vaccine can be achieved just by promoter design. This is an important finding, because NP modifications itself did not lead to improved antibody or T-cell responses in a previous study [56]. The combination of early and late promoter elements is well established for the expression of vaccine antigen in VACV- or MVA-based vaccines. Classical early-late promoters allow for a continuous antigen expression during the MVA replication cycle and a balanced immune response involving the activation of humoral and cellular immune responses [59]. This has been confirmed in several previous studies, when high levels of gene expression or protective immune responses induced by MVA have been obtained with synthetic promoters using different target antigens [28,60]. Thus, the local abundance of target proteins induced by synthetic promoters during early and late gene expression suggests enhanced cellular and humoral immune responses by improved presentation to effector cells locally and in lymph nodes [61]. In our study, as a proof of principle, we confirmed that using strong early elements or late-early combinations for the design of MVA-specific promoters represents a promising tool to further improve MVA vector-induced immunogenicity and efficacy.

When comparatively testing the expression of NP under the control of four newly designed late-early promoters and one strong early promoter, we did not detect obvious differences when using Western Blot analysis. Of note, future analysis using FACS analysis may contribute to characterizing the expression of NP driven by the different promoters in more detail. Our data are in line with previous data using the NP under the expression control of the well-established early-late promoter PmH5 [14]. In a mouse model for lethal IAV challenge infection, the different recombinant MVA-NP candidate vaccines improved the disease outcome when using prime-boost immunization strategies compared to control vaccinated mice. In the prime-boost vaccination schedule, we identified a MVA-PVFG-NP candidate vaccine to induce the lowest level of protection, since two out of the five mice died due to morbidity. Interestingly, we did not observe any differences in the immune responses after prime-boost vaccination and IAV-challenge infection, which could explain this reduced protective efficacy of MVA-PVGF-NP vaccination in this experimental setting. It will be of interest to further characterize the immune responses before challenge infection to identify more subtle differences in the quality and levels of immune responses that could be due to the IAV challenge infection itself. The impact of a challenge infection on the pre-existing antigen-specific immune responses has already been demonstrated in previous studies for IAV [62]. An important readout for vaccine-induced protection can be seen in the determination of viral load in the lungs. We did not analyze the viral load of the two MVA-PVGF-NP-vaccinated mice which succumbed to IAV infection. Thus, we cannot exclude that there were some effects of the MVA-PVGF-NP vaccination compared to control mice. At the end of the experiment, 28 days after infection, we did not detect virus in the lungs of the MVA-PLMU-NP-vaccinated mice. Here, it will be also helpful to further analyze the viral load at earlier time points to better compare the protective efficacy of these new promoter constructs with the one mediated by the PVGF-promoter. This effort might also further contribute to identifying immune correlates of protection. In this context, based on previous studies, the humoral immune responses are considered as a correlate of protection after prime-boost vaccination. In our previous studies, we confirmed that the strict early expression of vaccine antigens is associated with the more robust activation of a cellular immune response [38,63].

From this, we hypothesize that there are subtle differences in the quality and quantity of humoral immune responses after MVA-PVGF-NP vaccination which might account for the reduced outcome of protection after a prime-boost vaccination schedule. Of note, MVA-PVGF-NP-vaccinated mice mounted higher levels of NP antigen-specific T cells compared to the mice vaccinated with the MVA-PLMU-NP candidate vaccines. Since we also detected significantly higher levels of infectious virus in the lungs, a higher viral load in the lungs could be a possible explanation for the increased T cell response. A direct impact of higher virus loads on the magnitude of immune responses has been also reported by other studies on IAV and also on yellow fever virus [64,65]. Moreover, the earlier NP synthesis when under the transcriptional control pf PVGF also could have contributed to the increased activation of T cells. 

Interestingly, this less-efficient protective efficacy of the MVA-PVGF-NP candidate vaccine was blurred when we characterized the immunogenicity and efficacy of a single vaccination. In this prime-only vaccination schedule, the MVA-PVGF-NP and MVA-PLMU4-NP candidate vaccines proved to be beneficial in the activation of NP-specific T cells. Of note, protective efficacy after a single vaccination was also found with PLMU1-, PLMU2-, and PLMU3-promoter constructs. Following these vaccinations, we did not detect viral loads in the lungs at the end of the experiment and no obvious morbidity that necessitated euthanizing the mice before the end of the experiment. In contrast, high viral loads were detected in the lungs of the control mice. Since control mice had to be euthanized earlier in the experiment due to severe morbidity, it will be of interest to analyze the viral load in the lungs of MVA-NP-vaccinated mice at earlier time points to possibly detect differences and identify the mechanism of protection as mediated by the different MVA-NP-constructs. Previous studies have already confirmed that virus is present within the lungs of NP-vaccinated and challenged animals, indicating that there is no sterile immunity induced by the NP antigen. Such data may further support a role for T cells in mediating early protection. Interestingly, the robust activation of NP-specific antibodies was rapidly detectable in MVA-PVGF-NP- or MVA-PLMU4-NP-vaccinated mice. Moreover, the protective efficacy, as seen by morbidity and mortality, and also lung pathology, appeared to be significantly improved in MVA-PVGF-NP and MVA-PLMU4-NP candidate vaccines, with significantly reduced viral loads in the lungs compared to the control mice. Thus, MVA-PVGF-NP and MVA-PLMU4-NP elicited a protective immune response already at eight days post-challenge infection. At this time point, the immune response either overcomes IAV infection or the virally-induced lesions lead to the death of infected mice, as seen for the control mice. 

Thus, we hypothesize that viral load at earlier time points after challenge infection might also correlate with the detection of NP-specific CD8+ T cells in these mice. This is in line with previous studies where early vaccine-induced protection has been correlated with the activation of antigen-specific T cells [61]. This strong and early activation of T cells was further confirmed when we detected robust NP-specific CD8+ T cells after vaccination in the peripheral blood after single MVA-PVGF-NP or MVA-PLMU4-NP vaccination. From these data, we hypothesize that the protective efficacy of MVA-PVGF-NP or MVA-PLMU4-NP at such early time points is mainly mediated by CD8+ T cells. This has already been demonstrated in previous studies, where robust NP-specific CD8+ T cells were associated with improved protection against lethal IAV challenge infection in mice [14,55,66,67]. Of note, to further investigate the pattern of NP-epitope-specific CD8+ T cells in peripheral blood, additional experiments are useful to analyze these CD8+ T cell populations in the spleen simultaneously. Indeed, the depletion of CD8+ T cells completely abrogated the robust MVA-PVGF-NP-induced protection and reduced the protective efficacy of MVA-PLMU4-NP, emphasizing the potential of these newly designed promoters to strongly activate T cell responses. Our data from the CD8+ T cell depletion experiments indicated a tendency towards a slightly reduced disease severity, as seen by the viral load, pathology, and morbidity in the MVA-PLMU4-NP-vaccinated and CD8+-depleted mice. These data indicate that MVA-PLMU4-NP vaccination resulted in a different activation of immune responses that are not completely abrogated by the CD8+ T cell depletion. This might be best explained by the late gene expression from the PLMU4-promoter, which could also drive a stronger activation of other components of the immune system. In this context, it will be interesting to also evaluate the possible role of CD4+ T cells in mediating such a partial vaccine-induced protection against lethal IAV challenge infection. Moreover, the role of NP-IgG binding antibodies could be also analyzed in more detail in future studies. High levels of anti-NP antibodies induced by MVA-PLMU4-NP may have contributed to vaccine-induced protection. This is supported by previous studies which demonstrate the protective capacity of anti-NP IgG [57,58]. This is surprising because NP, an internal structural protein produced within the infected cell, should remain largely undetected by the B cell arm of the immune system. 

The involvement of antibody-dependent, cell-mediated cytotoxicity (ADCC) with the secretion and/or cell surface presentation of NP has been proposed, but the exact mechanisms of protection correlating with the antiviral activity of anti-NP IgG remain unclear [57]. In this context, it will be also of interest to characterize the levels of virus-neutralizing antibodies after IAV-challenge infection. A potential role for IAV-neutralizing antibodies has been already confirmed by a different experimental approach.

The robust activation of T cells is a promising approach to overcome the limitations of vaccines. In addition to an important role for antibodies, CD4+ and also CD8+ T cells might then also contribute to improved vaccine-induced protection. This is especially true for licensed IAV vaccines which have to be updated every season due to the emergence of new IAV strains. In this context, the activation of T cells and, more precisely, the activation of NP-specific T cells has been considered to induce more cross-reactive protection, as confirmed in previous studies [68]. In this context, it will be of interest to also evaluate the capacity of PVGF-promoter construct and PLMU4-promoter construct to mediate protection against other IAV strains. 

A recombinant MVA expressing NP and M1 as target antigens has already been employed in human trials, leading to T- cell expansion and partial protection after IAV challenge [69,70]. This vaccine uses the natural early-late promoter P7.5 to deliver NP. P7.5 is a well-characterized promoter of moderate late transcriptional activity which is capable of inducing solid T cell responses. However, experiments show that the antigen delivery by MVA and the consequent immune response can be enhanced by using the stronger synthetic early-late promoter PmH5 [61]. Thus, our approach, to further enhance this NP specific T cell response by the use of new composite promoters for the expression of NP in MVA-based candidate vaccines proved to be promising and should be analyzed in more detail in future studies which should also involve challenge infection with different IAVs and use other animal models. Moreover, the use of these new promoters might also be a promising approach against other pathogens, e.g., betacoronaviruses, where a strong T cell response might be the key for achieving robust vaccine induced crossed-protection. 

## 5. Conclusions

In summary, our data indicate a beneficial effect of the use of the PVGF- and the PLMU4-promoters on the immunogenicity and protective efficacy of a MVA-NP candidate vaccine in a lethal IAV mouse model. However, since we did not detect a direct correlation between the outcome of protection and the activation of specific immune responses. neither for the PVGF-promoter construct nor for the PLMU4-promoter construct, future, more detailed studies have to be undertaken which also evaluate more and different time-points before and after challenge infection. In this context, follow-up experiments will be required to analyze the immunogenicity of the different MVA-NP candidate vaccines prior to the challenge in more detail. Such future studies should also analyze the viral load in the organs at the same time post-challenge infection. These data will further contribute to correlating specific immune responses with the outcome of protection as induced by the different MVA-NP candidate vaccines. Moreover, future studies, also using other pathogens, will be important to further characterize the promoter effect on vaccine-induced protection in more detail. 

## Figures and Tables

**Figure 1 pathogens-12-00867-f001:**
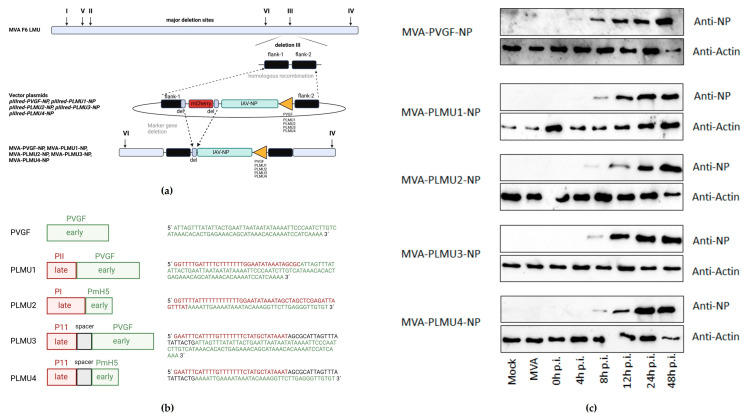
(**a**) Schematic diagram of the MVA genome with the major deletion sites I through VI. The site of deletion III was targeted for insertion of the gene sequence encoding the nucleoprotein (NP) of influenza A virus strain A/Puerto Rico/8/1934/H1N1 (IAV). IAV-NP was placed under transcriptional control of either the natural early vaccinia virus promoter PVGF or the synthetic chimeric promoters PLMU1, PLMU2, PLMU3, and PLMU4 within the MVA vector plasmid pIIIred-PGVF-NP, pIIIred-PLMU1-NP, pIIIred-PLMU2-NP, pIIIred-PLMU3-NP, or pIIIred-PLMU4-NP. Insertion occurred via homologous recombination between MVA DNA sequences (flank-1 and flank-2) adjacent to deletion site III in the MVA genome and copies cloned in the vector plasmid. MVA-PVGF-NP, MVA-PLMU1-NP, MVA-PLMU2-NP, MVA-PLMU3-NP, and MVA-PLMU4-NP were clonally isolated by plaque purification screening for co-production of the red fluorescent marker protein mCherry. A repetition of short flank-1-derived DNA sequences (del) served to remove the marker gene by intragenomic homologous recombination (marker gene deletion). Created with BioRender.com. (**b**) Overview of chimeric promoter design. Created with BioRender.com. (**c**) Synthesis of full-length IAV-NP. CEF cells were infected at a MOI of 1 and cell lysates were prepared at the indicated time-points. Polypeptides in cell lysates were separated by SDS-PAGE and analyzed with a monoclonal antibody directed against the nucleoprotein (1:100). Created with BioRender.com.

**Figure 2 pathogens-12-00867-f002:**
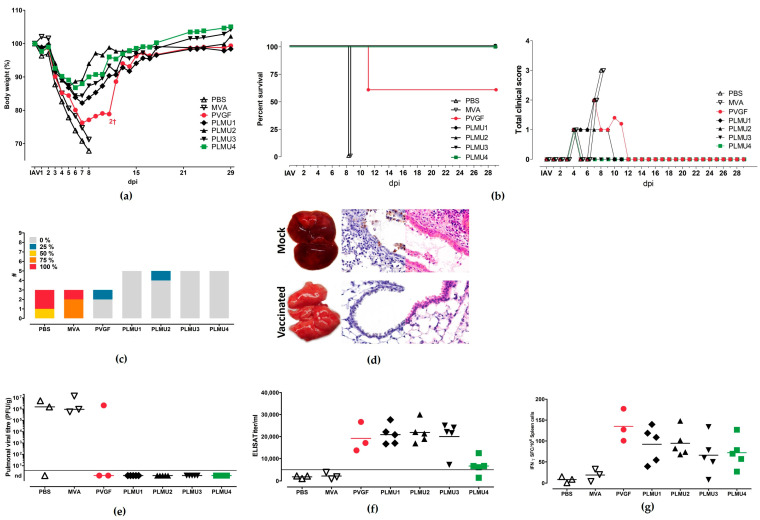
Protective capacity of MVA-PVGF-NP (PVGF, n = 5), MVA-PLMU1-NP (PLMU1, n = 5), MVA-PLMU2-NP (PLMU2, n = 5), MVA-PLMU3-NP (PLMU3, n = 5), or MVA-PLMU4-NP (PLMU4, n = 5) prime-boost immunization against A/Puerto Rico/8/1934/H1N1 infection in C57BL/6 mice. Groups of mice were immunized twice over a 21-day interval with 10^8^ pfu MVA-PVGF-NP, MVA-PLMU1-NP, MVA-PLMU2-NP, MVA-PLMU3-NP, or MVA-PLMU4-NP. Mice immunized with non-recombinant MVA (MVA, n = 3) or saline (PBS, n = 3) served as controls. Four weeks post-booster immunization, mice were infected intranasally with a lethal dose of A/Puerto Rico/8/1934/H1N1 and were monitored for (**a**) body weight changes and (**b**) behaviour and general condition in clinical scores. Mice were euthanized 28 days after the challenge or after humane endpoints were reached. At necropsy, (**c**) percent of lung affected by pneumonia was determined, (**d**) pulmonal macroscopic and histologic lesions were recorded in routine stain and immunohistochemistry (displaying intact and necrotic epithelia with Influenza antigen), and, additionally, (**e**) pulmonary viral loads, (**f**) serum IgG antibodies, and (**g**) IFN-γ+ spot-forming cells (SFC) were analyzed by ELISPOT.

**Figure 3 pathogens-12-00867-f003:**
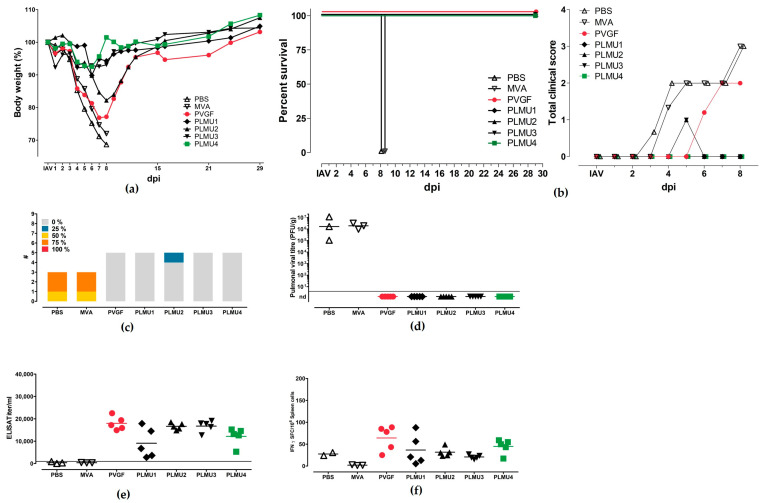
Protective capacity of MVA-PVGF-NP (PVGF, n = 5), MVA-PLMU1-NP (PLMU1, n = 5), MVA-PLMU2-NP (PLMU2, n = 5), MVA-PLMU3-NP (PLMU3, n = 5), or MVA-PLMU4-NP (PLMU4, n = 5) prime immunization against A/Puerto Rico/8/1934/H1N1 infection in C57BL/6 mice. Groups of mice were immunized once with 10^8^ pfu MVA-PVGF-NP, MVA-PLMU1-NP, MVA-PLMU2-NP, MVA-PLMU3-NP, or MVA-PLMU4-NP. Mice immunized with non-recombinant MVA (MVA, n = 3) or saline (PBS, n = 3) served as controls. After 28 d.p.v., mice were infected intranasally with a lethal dose of A/Puerto Rico/8/1934/H1N1 and mice were monitored for (**a**) body weight changes and (**b**) behaviour and general condition in clinical scores. Mice were euthanized 28 days after the challenge infection or after humane endpoints were reached. At necropsy, (**c**) percent affected by pneumonia was determined, and, additionally, (**d**) pulmonary viral loads, (**e**) serum IgG antibodies, and (**f**) IFN-γ+ spot-forming cells (SFC) were analyzed by ELISPOT.

**Figure 4 pathogens-12-00867-f004:**
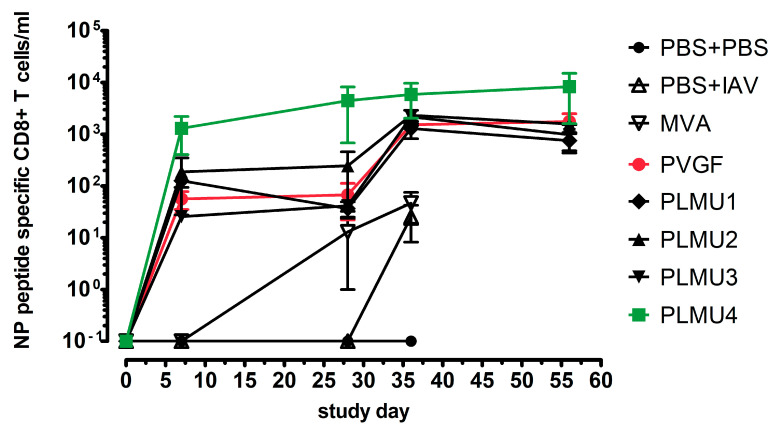
CD8+ T cell expansion in MVA−NP−primed C57BL/6 mice. Groups of mice (n = 2–5) were immunized once with 10^8^ pfu MVA-PVGF-NP (n = 5 per group), MVA-PLMU1-NP, MVA-PLMU2-NP, MVA-PLMU3-NP, or MVA-PLMU4-NP (n = 5 per group). Mice immunized with non-recombinant MVA (MVA, n = 3) or saline without (PBS, n = 2) and with challenge (PBS + IAV, n = 3) served as controls. At w4 post-immunization, mice were infected intranasally with a lethal dose of A/Puerto Rico/8/1934/H1N1 and peripheral blood was tested at the indicated time points for IAV-NP-specific CD8+ T-cell expansion.

**Figure 5 pathogens-12-00867-f005:**
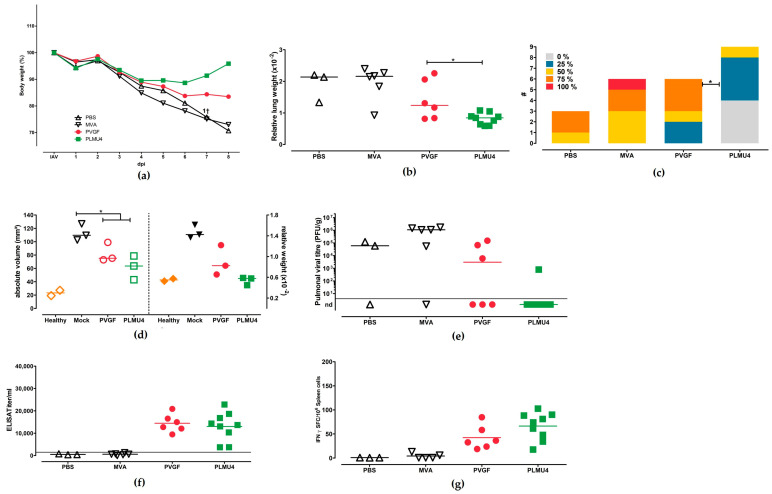
Protective capacity of MVA−PVGF−NP (PVGF, n = 6) or MVA−PLMU4−NP (PLMU4, n = 9) prime immunization against A/Puerto Rico/8/1934/H1N1 infection in C57BL/6 mice. Groups of mice were immunized once with 10^8^ pfu MVA-PVGF-NP or MVA-PLMU4-NP. Mice immunized with non-recombinant MVA (MVA, n = 6) or saline (PBS, n = 3) served as controls. Then, 28 d.p.v., mice were infected with a lethal dose of A/Puerto Rico/8/1934/H1N1 and were monitored for (**a**) body weight changes. End point of the study was eight days post-challenge. At necropsy, (**b**,**c**) relative lung weights and percent affected by pneumonia, as well as (**d**) quantitative stereological estimation of lung volume affected by pneumonia, were determined; additionally, (**e**) pulmonary viral loads, (**f**) serum IgG antibodies, and (**g**) IFN-γ+ spot-forming cells (SFC) were analyzed. Data were analyzed by Mann–Whitney test unless otherwise indicated. Asterisks represent statistically significant differences: * *p* < 0.05.

**Figure 6 pathogens-12-00867-f006:**
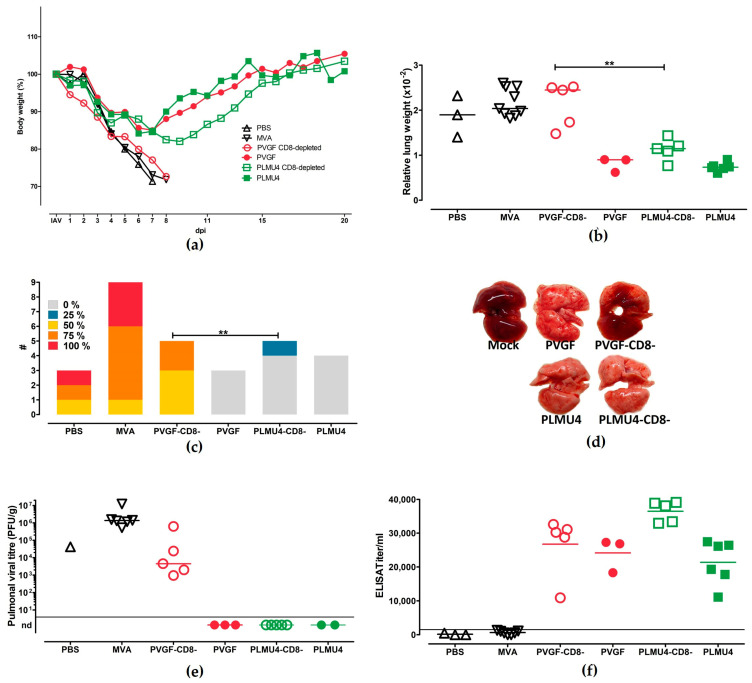
Protective capacity of MVA−PVGF−NP (PVGF, n = 3), or MVA−PLMU4−NP (PLMU4, n = 6) prime immunization against A/Puerto Rico/8/1934/H1N1 infection in CD8+ T cell-depleted C57BL/6 mice (combination of two experiments). Groups of mice were immunized once with 10^8^ pfu MVA-PVGF-NP or MVA-PLMU4-NP. Mice immunized with non-recombinant MVA (MVA, n = 9) or saline (PBS, n = 3) served as controls. Prior to challenge infection with A/Puerto Rico/8/1934/H1N1 at 28 d.p.v., CD8+ T cells were depleted by intraperitoneal injection of an anti-CD8+ antibody in MVA-PVGF-NP- and MVA-PLMU4-NP-vaccinated mice (n = 5, each). (**a**) Body weight was monitored. Mice were euthanized 28 days after challenge or after humane end points were reached. At necropsy, (**b**–**d**) relative lung weight and percent affected by pneumonia were determined and, additionally, (**e**) pulmonary viral loads and (**f**) serum IgG antibodies were analyzed. Data were analyzed by Mann–Whitney test unless otherwise indicated. Asterisks represent statistically significant differences between two groups: ** *p* < 0.01.

## Data Availability

Not applicable.

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
