# Peer review of "Newly Designed Poxviral Promoters to Improve Immunogenicity and Efficacy of MVA-NP Candidate Vaccines against Lethal Influenza Virus Infection in Mice"

_pathogens, 2023, doi:10.3390/pathogens12070867_

Round 1
Reviewer 1 Report
This article is well-written and provides sufficient detail for the objectives and materials/methods descriptions.
I have concerns about the study designs and supportive results used to reach the overall conclusion that newly designed poxviral promoters are improvements vs the natural early promoter (PVGF):
Assessment of the immune responses are limited to pre-challenge time points with the exception of the CD8 T cell responses shown in Fig. 4. This makes it difficult to discern how much the virus infection is impacting the measured immune responses.
The immune response and viral load/pathology parameters are assessed at different time points for the naive control vs immunized mice for the first two experiments (Fig. 2 and 3). There is little value in comparing viral loads and histopathology among groups when samples are collected on Day 8-11 post-challenge vs 4 weeks later. The last experiment (Fig. 5) does include sampling on Day 8 post-challenge but limits the assessment to only a single synthetic promoter construct (PLMU4) and the results are not overly convincing (one mouse out of 3 in the PBS control group tests negative for infectious virus). The CD8 T cell responses (g), lung weight (b) and lung volume (d) in the PLMU4 group are relatively similar to the PVGF group, making it difficult to convincingly argue that the PLMU4 promoter is an improvement vs the natural promoter.
I have further concerns regarding the number of animals used to demonstrate that the CD8 T cell responses induced by the PLMU4 expressing MVA are superior vs the other synthetic and the natural promoter (Fig. 4). The legend for Fig. 4 shows that 2-5 mice were used for measuring the CD8 T cell responses in peripheral blood, but it is not clear how many mice were included for each group. The error bars for the PLMU4 results are very large, making it difficult to evaluate the overall conclusion of superiority by the PLMU4 promoter.
Minor issues include the following:
The title of the x-axis for Fig. 4 is "dpi". This should be changed to Study Day (or something similar) since the initial data points represents pre-challenge time points.
The graphs for the clinical scores are difficult to understand and the format is slightly different between Fig. 2 and 3. For Fig. 2, the symbols for the control groups are not present/visible but one would assume that these developed high scores. This should be clarified. Also, a Kaplan-Meyer survival plot for Fig. 2 and 3 would be helpful in comparing the results among control and treatment groups.
There appears to be missing data from the two mice that succumbed to the infection for Fig. 2f, 2g, and 2h (only 3 of 5 mice are represented). Were these mice found dead and therefore not assessed for viral loads, ELISA and ELISpot at time of necropsy? For the ELISpot data, the PVGF group appears to have improved responses vs the synthetic promoter constructs, but this is not addressed/discussed. Could this potentially be due to increased acute viral loads for this group.....?
The lethal dose is not mentioned. How much does the 10^3 TCID50 dose correspond to? 10x LD50?
The antibody responses are limited to binding Ab (ELISA). It would have improved the analysis if neutralizing antibody levels had been measured as well.
The article is generally well-written with only minor grammatical errors. For example, line 95 (...for expressing of...).
Author Response
Point-by-Point response to the reviewers`comments [MS# 2412352]
We thank the reviewers for their insightful observations and comments, which have all been answered and enabled us to resubmit an improved manuscript. Below we listed all reviewers’ points in black and our answers are in blue. Within the manuscript, edits in response to the reviewers’ comments are highlighted in yellow.
Reviewer Comments:
Reviewer 1:
This article is well-written and provides sufficient detail for the objectives and materials/methods descriptions.
We greatly appreciate the reviewer’s overall positive and very constructive response which enabled us to submit a clearly improved manuscript.
I have concerns about the study designs and supportive results used to reach the overall conclusion that newly designed poxviral promoters are improvements vs the natural early promoter (PVGF):
We totally agree with the reviewer that the data presented here do not definitely support a clear advantage of the PLMU4-promoter construct. To support this, more detailed studies have to be undertaken as now it is also discussed within the new discussion part. In addition, we now also included the PVGF as promising promoter construct within the conclusion section (see lines 739-747:”In summary, our data indicate a beneficial effect of the use of the PVFG-and the PLMU4 promoter on the immunogenicity and protective efficacy of a MVA-NP-candidate vaccine in a lethal IAV mouse model. However, since we did not detect a direct correlation between the outcome of protection and the activation of specific immune responses neither for the PVGF-promoter construct nor for the PLMU4-promoter construct, future more detailed studies have to be undertaken also evaluating more and different time-points before and after challenge infection. These data will further contribute to correlate specific immune responses with the outcome of protection as induced by the different MVA-NP-candidate vaccines”).
Assessment of the immune responses are limited to pre-challenge time points with the exception of the CD8 T cell responses shown in Fig. 4. This makes it difficult to discern how much the virus infection is impacting the measured immune responses.
So far our analyses on the immune responses except of Figure 4 are limited to post-challenge time points which relates to the same concern as addressed by the reviewer. To refer to the likely impact of the challenge infection on the preexisting immune responses, we now included several statements within the discussion section (see lines 621-625:“It will be of interest to further characterize the immune responses before challenge infection to identify more subtle differences in the quality and levels of immune responses that could be due to the IAV challenge infection itself. The impact of a challenge infection on the pre-existing antigen-specific immune responses has been already demonstrated in previous studies for IAV“, lines 741-747:“ However, since we did not detect a direct correlation between the outcome of protection and the activation of specific immune responses neither for the PVGF-promoter construct nor for the PLMU4-promoter construct, future more detailed studies have to be undertaken also evaluating more and different time-points before and after challenge infection. These data will further contribute to correlate specific immune responses with the outcome of protection as induced by the different MVA-NP-candidate vaccines“).
The immune response and viral load/pathology parameters are assessed at different time points for the naive control vs immunized mice for the first two experiments (Fig. 2 and 3). There is little value in comparing viral loads and histopathology among groups when samples are collected on Day 8-11 post-challenge vs 4 weeks later. The last experiment (Fig. 5) does include sampling on Day 8 post-challenge but limits the assessment to only a single synthetic promoter construct (PLMU4) and the results are not overly convincing (one mouse out of 3 in the PBS control group tests negative for infectious virus). The CD8 T cell responses (g), lung weight (b) and lung volume (d) in the PLMU4 group are relatively similar to the PVGF group, making it difficult to convincingly argue that the PLMU4 promoter is an improvement vs the natural promoter.
We agree that the comparison of the selected parameters at different time points is suboptimal to assess the subtle beneficial effects of different candidate vaccines. However, since the repetition of experiments evaluating the effects of the MVA-NP candidate vaccines for all experimental groups at earlier time points was not in the scope of this revision, we included this limitation within the discussion section (see lines 621-633:” It it will be of interest to further characterize the immune responses before challenge infection to identify more subtle differences in the quality and levels of immune responses that could be due to the IAV challenge infection itself. The impact of a challenge infection on the pre-existing antigen-specific immune responses has been already demonstrated in previous studies for IAV. An important readout for vaccine induced protection is the determination of viral load in the lungs. We did not analyze the viral load of the two MVA-PVGF-NP vaccinated mice succumbing to IAV infection. Thus, we cannot exclude that there were some effects of the MVA-PVGF-NP vaccination compared to control mice. At the end of the experiment 28 days after infection we did not detect virus in the lungs of MVA-PLMU-NP vaccinated mice. Here, it will be also helpful to further analyze the viral load at earlier time points to better compare the protective efficacy of these new promoter constructs with the one mediated by the PVGF-promoter. This effort might also further contribute to identify immune correlates of protection”). Moreover, we also removed the data from lung weight from Figure 2 and Figure 3, since this does not give much information when measured at such different time points.
I have further concerns regarding the number of animals used to demonstrate that the CD8 T cell responses induced by the PLMU4 expressing MVA are superior vs the other synthetic and the natural promoter (Fig. 4). The legend for Fig. 4 shows that 2-5 mice were used for measuring the CD8 T cell responses in peripheral blood, but it is not clear how many mice were included for each group. The error bars for the PLMU4 results are very large, making it difficult to evaluate the overall conclusion of superiority by the PLMU4 promoter.
An important readout for vaccine induced protection is the determination of viral load in the lungs. We did not analyze the viral load of the two MVA-PVGF-NP vaccinated mice succumbing to IAV infection. Thus, we cannot exclude that there were some effects of the MVA-PVGF-NP vaccination compared to control mice. At the end of the experiment 28 days after infection we did not detect virus in the lungs of MVA-PLMU-NP vaccinated mice. Here, it will be also helpful to further analyze the viral load at earlier time points to better compare the protective efficacy of these new promoter constructs with the one mediated by the PVGF-promoter. This effort might also further contribute to identify immune correlates of protection
Minor issues include the following:
The title of the x-axis for Fig. 4 is "dpi". This should be changed to Study Day (or something similar) since the initial data points represents pre-challenge time points.
We thank the reviewer for the excellent suggestion. We changed the title of the x-axis to Study day, which much better expresses the study design and the sampling time poins (see Figure 4).
The graphs for the clinical scores are difficult to understand and the format is slightly different between Fig. 2 and 3. For Fig. 2, the symbols for the control groups are not present/visible but one would assume that these developed high scores. This should be clarified. Also, a Kaplan-Meyer survival plot for Fig. 2 and 3 would be helpful in comparing the results among control and treatment groups.
We highly appreciate the great suggestion. We optimized the graphs for clinical scores to be more easily to understand. For this we also included details about score sheet in the Material and Method section to identify the different clinical disease symptoms that have been counted for the clinical score as presented in the graphs. Moreover, we also included Kaplan-Meyer survival plots for Fig.2 and Fig.3.
There appears to be missing data from the two mice that succumbed to the infection for Fig. 2f, 2g, and 2h (only 3 of 5 mice are represented). Were these mice found dead and therefore not assessed for viral loads, ELISA and ELISpot at time of necropsy? For the ELISpot data, the PVGF group appears to have improved responses vs the synthetic promoter constructs, but this is not addressed/discussed. Could this potentially be due to increased acute viral loads for this group.....?
Yes, the two MVA-PVFG-NP vaccinated mice that succumbed to the infection were found dead. Thus, we did not include these mice for characterization of viral loads and immune responses. We have clarified and corrected this in the manuscript and also included a section within the results and discussion sections (see lines 367:“ two mice from the MVA-PVGF-NP vaccinated group which had to be euthanizeddied at day 11 after challenge“, see line 625-629: “An important readout for vaccine induced protection is seen by the determination of viral load in the lungs. We did not analyze the viral load of the two MVA-PVGF-NP vaccinated mice succumbing to IAV infection. Thus, we cannot exclude that there were some effects of the MVA-PVGF-NP vaccination compared to control mice“).
We agree with the Reviewer that the MVA-PVGF-NP vaccinated group appears to show enhanced levels of NP-specific T cells compared to the other groups as measured by ELISPOT assay. In response to this suggestion, we have revised the results section of the manuscript to better describe the enhanced cellular immune response upon MVA-PVGF-NP vaccination. We also included a paragraph within the discussion section to address the potential mechanisms of an improved activation of T cells. Here we also discussed the possible impact of an increased viral load for the activation of immune responses (see lines 640-648:” Of note, MVA-PVGF-NP vaccinated mice mounted higher levels of NP antigen- specific T cells compared to the mice vaccinated with the MVA-PLMU-NP candidate vaccines. Since we also detected significantly higher levels of infectious virus in the lungs. A higher viral load in the lung could be a possible explanation for the increased T cell re-sponse. A direct impact of higher virus loads on the magnitude of immune responses had been also reported from other studies with IAV and also with SARS-CoV-2. Moreover, the earlier NP synthesis when under transcriptional control pf PVGF could have also contributed to the increased activation of T cells”).
The lethal dose is not mentioned. How much does the 10^3 TCID50 dose correspond to? 10x LD50?
Due to animal protection requirements, we are not titrating the lethal dose 50 of stock virus preparations used in challenge experiments itself. We established our lethal influenza—mouse model using the mouse-adapted influenza A virus (A/Puerto Rico/8/1934/H1N1) and confirmed the development of lethal disease as shown by weight loss and clinical disease symptoms as monitored in score sheet. The dosage of 103 TCID50 was used for challenge since this infection reproducibly resulted in disease development fatal outcome in >90% of the mice . To make this more clear, we included a sentence within the Material and Methods section (see ln 189-190: “A dose of 103 TCID50 was used since this infection reproducibly resulted in fatal disease in >90% of the mice “).
The antibody responses are limited to binding Ab (ELISA). It would have improved the analysis if neutralizing antibody levels had been measured as well.
We agree with the Reviewer that the measurement of neutralizing antibodies after challenge infection would have improved the analysis. However, since we aimed in correlating the immune responses as induced by the recombinant MVA-NP candidate vaccines with the outcome of protection, we focused on the determination of NP-binding antibodies. To consider this suggestion by Reviewer 1, we included a section within the discussion elaborating the contribution of neutralizing antibodies to the protective efficacy (see lines 697-700: ”In this context, it will be also of interest to characterize the levels of virus-neutralizing antibodies after IAV-challenge infection. A potential role for post-challenge IAV-neutralizing antibodies has been already confirmed from different experimental approaches.”).

Reviewer 2 Report
In the manuscript titled "Newly designed poxviral promoters to improve immunogenicity and efficacy of MVA-NP candidate vaccines against lethal Influenza virus infection in mice," submitted by Langenmayer and colleagues, the authors investigate the efficacy of MVA-based vaccines expressing the nucleoprotein (NP) of IAV under the control of five different promoters. The study explores the induction of humoral and cellular immune responses, as well as the protective efficacy of the vaccines following one or two prophylactic immunizations. The authors successfully detect NP-specific antibodies and CD8 T cells and find that an MVA recombinant expressing NP under the control of the artificial promoter PLMU4 elicits the most effective protection.
The manuscript is well-written, and the figures are visually appealing. The main thread of the study is easy to follow, and overall, the manuscript is acceptable. However, there are a few areas where clarity and improvements are desired. I would like to highlight the following points that should be addressed:
i. It would be beneficial to include Figure S1A in the main text. The sequence information of the different promoters would be of interest to the readers.
ii. It is recommended that the authors employ a more meaningful method, such as flow cytometry, to test the antigen expression strength from the different promoters at the single-cell level. This would provide a more accurate determination of antigen expression.
iii. Further explanation is needed in the figure legend of Figure 2e, particularly regarding the identification of macrophages and cells expressing the NP antigen. As a reader unfamiliar with IHC, it is difficult to understand the cell types involved.
iv. Consider conducting a correlation analysis between NP-specific antibody levels and T-cell frequencies for Figures 3 and 4. Is there any correlation between AB-titer or T-cell frequency and clinical outcome? While it is assumed this may not be the case, it should be addressed and could improve the discussion.
v. For Figure 4, it is recommended to use a logarithmic scale or break the Y-axis for improved clarity.
vi. The substantial difference in T-cell frequencies between blood and spleen, as well as the differing outcomes observed with different promoters, should be investigated in a single experiment. It is crucial to analyze T cells in both blood and spleen simultaneously. This discrepancy is a significant concern, especially since T cells in blood were not determined or shown in Figure 5.
vii. The statement in line 578 contradicts Figure 1b, where PVGF-expressed NP appears to be expressed more strongly and earlier than NP expressed under the control of artificial promoters. This discrepancy should be addressed in the discussion.
viii. The statement in line 585 seems isolated from the findings of Figure 3, where a correlation between T cells and clinical outcome is not expected. This discrepancy needs further discussion.
ix. Figure 6 indicates that CD8 T cells may not be the only correlate to clinical outcome. In light of the absence of a correlation between NP-specific antibody levels and clinical outcome, the argumentation is not convincing. The authors should further address and discuss factors such as the role of CD4 T cells, recruitment of immune cells to the site of infection, or the dynamics of adaptive immunity.
Additionally, I have identified some minor errors related to editing:
- Line 437: Reference to Figure 4 is missing.
- Line 513: The figure reference needs to be corrected to Figures 6a,b.
- Please maintain consistency with the time units, sticking to either "days post immunization/infection" or "weeks" throughout the manuscript (e.g., line 388-391 and line 429).
Overall, addressing these points and minor editing errors will significantly enhance the clarity and quality of the manuscript.
Author Response
Point-by-Point response to the reviewers`comments [MS# 2412352]
We thank the reviewers for their insightful observations and comments, which have all been answered and enabled us to resubmit an improved manuscript. Below we listed all reviewers’ points in black and our answers are in blue. Within the manuscript, edits in response to the reviewers’ comments are highlighted in yellow.
Reviewer 2:
In the manuscript titled "Newly designed poxviral promoters to improve immunogenicity and efficacy of MVA-NP candidate vaccines against lethal Influenza virus infection in mice," submitted by Langenmayer and colleagues, the authors investigate the efficacy of MVA-based vaccines expressing the nucleoprotein (NP) of IAV under the control of five different promoters. The study explores the induction of humoral and cellular immune responses, as well as the protective efficacy of the vaccines following one or two prophylactic immunizations. The authors successfully detect NP-specific antibodies and CD8 T cells and find that an MVA recombinant expressing NP under the control of the artificial promoter PLMU4 elicits the most effective protection.
The manuscript is well-written, and the figures are visually appealing. The main thread of the study is easy to follow, and overall, the manuscript is acceptable. However, there are a few areas where clarity and improvements are desired. I would like to highlight the following points that should be addressed:
We greatly appreciate the reviewer’s positive response and for pointing out the specific points which enabled us to submit a clearly improved manuscript. We tried to corroborate our findings by addressing these points in more detail within the discussion section of the revised manuscript.
- It would be beneficial to include Figure S1A in the main text. The sequence information of the different promoters would be of interest to the readers.
This was a good suggestion. Based on the Reviewers comment, we included Figure S1A as Figure 1B within the main text of the manuscript.
- It is recommended that the authors employ a more meaningful method, such as flow cytometry, to test the antigen expression strength from the different promoters at the single-cell level. This would provide a more accurate determination of antigen expression.
We agree with the Reviewer that flow cytometry would provide another very meaningful method to characterize the expression strength of the different promoters at the single-cell level. However, so far we have not established a proper protocol to comparatively monitor the NP expression of MVA-NP infected CEF by FACS. Due to the limited time for revision of the manuscript, a meaningful determination of the NP-antigen expression is out of the scope of this study. We included a sentence within the discussion to highlight that further methods might provide a more detailed insight (see lines 608-612: ”When comparatively testing the expression of NP under the control of four newly de-signed late-early promoters and one strong early promoter, we did not detect obvious differences when using Western Blot analysis. Of note, future analysis using FACS analysis may contribute to characterize the expression of NP driven by the different promoters in more detail”).
iii. Further explanation is needed in the figure legend of Figure 2e, particularly regarding the identification of macrophages and cells expressing the NP antigen. As a reader unfamiliar with IHC, it is difficult to understand the cell types involved.
This was an excellent suggestion. We included a more detailed figure legend to Figure 2e to better demonstrate the different cell types included within the IHC staining. In addition, we also included a more specific description within the text (see lines 356-357:” Immunohistochemistry detected NP antigen in intact and necrotic epithelia (Figure 2d), as well as in macrophages and alveolar septa (data not shown)”).
- Consider conducting a correlation analysis between NP-specific antibody levels and T-cell frequencies for Figures 3 and 4. Is there any correlation between AB-titer or T-cell frequency and clinical outcome? While it is assumed this may not be the case, it should be addressed and could improve the discussion.
As also indicated within the discussion section, there was no clear correlation between the clinical outcome, the T cell levels and the antibody titers as activated by the different recombinant MVA-NP vaccine candidates. Future studies are required to more closely analyse the immune responses (T Cells and antibodies) in more detail also including additional time points. In addition, such future studies would also include a more detailed analysis of viral load in the lungs also comparing more time points. Such studies should especially focus on different time points early after infection and later time points. Moreover, in protected animals, a very late time point for analysis should also be included to characterize the impact of memory responses for protective efficacy. To make this more clear, we included a section within the discussion (see lines 741-747: ”However, since we did not detect a direct correlation between the outcome of protection and the activation of specific immune response neither for the PVGF-promoter construct nor for the PLMU4-promoter construct, future more detailed studies have to be undertaken also evaluating more and different time-points before and after challenge infection. These data will further contribute to correlate specific immune responses with the outcome of protection as induced by the different MVA-NP-candidate vaccines”).
- For Figure 4, it is recommended to use a logarithmic scale or break the Y-axis for improved clarity.
This was a good suggestion. Based on the Reviewers comment we modified figures and improved clarity.
- The substantial difference in T-cell frequencies between blood and spleen, as well as the differing outcomes observed with different promoters, should be investigated in a single experiment. It is crucial to analyze T cells in both blood and spleen simultaneously. This discrepancy is a significant concern, especially since T cells in blood were not determined or shown in Figure 5.
We agree with the Reviewer that it is crucial to simultaneously analyze T cell responses in both blood and spleen. Thus additional experiments evaluating the T cells in blood or in spleen would significantly improve the outcome of the results. Since such an experiment would require a substantial number of additional animals and at least 8 weeks of time, we could not include data from such an experiment. We included a section within the discussion to address this limitation of our study (see lines 660-662:”Of note, to further investigate the pattern of NP-epitope specific CD8+ T cells in peripheral blood, additional experiments are useful to analyse these CD8+ T cell populations in the spleen simultaneously”).
vii. The statement in line 578 contradicts Figure 1b, where PVGF-expressed NP appears to be expressed more strongly and earlier than NP expressed under the control of artificial promoters. This discrepancy should be addressed in the discussion.
This is an excellent suggestion. We included a section in the discussion. In addition, we also included a sentence in the results section to better address the point raised by reviewer 2 (see lines 650-652: ”Moreover, the earlier NP synthesis under transcriptional control of PVGF as shown in Western Blot analysis could have also contributed to the increased activation of NP-specific T cells”).
viii. The statement in line 585 seems isolated from the findings of Figure 3, where a correlation between T cells and clinical outcome is not expected. This discrepancy needs further discussion.
This is a good suggestion. We agree with the Reviewer that the discrepancy needs further discussion. For this, we included a section within the discussion (see lines 656-668:”Of note, protective efficacy after a single vaccination was also found with PLMU1-, PLMU2- and PLMU3-promoter constructs. Following these vaccinations, we did not detect viral loads in the lungs at the end of the experiment and no obvious morbidity that afforded to euthanize the mice before the end of the experiment. In contrast, high viral loads have been detected in lungs of control mice. Since control mice had to be euthanized already earlier in the experiment due to severe morbidity, it will be of interest to analyze the viral load in the lungs of MVA-NP vaccinated mice at earlier time points to possibly detect differences and identify the mechanism of protection as mediated by the different MVA-NP-constructs. Previous studies already confirmed that virus is present within the lungs NP-vaccinated and challenged animals indicating that there is no sterile immunity induced by NP antigen. Such data may further support a role for T cells to mediate early protection”).
- Figure 6 indicates that CD8 T cells may not be the only correlate to clinical outcome. In light of the absence of a correlation between NP-specific antibody levels and clinical outcome, the argumentation is not convincing. The authors should further address and discuss factors such as the role of CD4 T cells, recruitment of immune cells to the site of infection, or the dynamics of adaptive immunity.
We agree with the Reviewer that the data presented within our manuscript indicate that CD8+ T cells may not be the only correlate of vaccine induced protection. Our data from the T cell depletion using a CD8+ specific antibodies only indicated that CD8+ T cells are essentially required to mediate the vaccine induced protection. To make this more clear we revised the discussion section and indicated the impact of CD8+ T cells based on our results and emphasized that also other components of adaptive immunity e.g. CD4+ T cells and/or antibodies might play a role (see lines 692-702:”Our data from the CD8+ T cell depletion experiments indicated a tendency of a slightly reduced disease severity as seen by viral load, pathology and morbidity in the MVA-PLMU4-NP vaccinated and CD8+ depleted mice. These data indicated that MVA-PLMU4-NP vaccination resulted in a different activation of immune responses that are not completely abrogated by the CD8+ T cell depletion. This might be best explained by the late gene expression from the PLMU4-promoter which could also drive a stronger activation of other components of the immune system. In this context, it will be interesting to also evaluate the possible role of CD4+ T cells for mediating such a partial vaccine induced protection against lethal IAV challenge infection. Moreover, the role of NP-IgG binding antibodies could be also analyzed in more detail in future studies”).
Comments on the Quality of English Language
Additionally, I have identified some minor errors related to editing:
Line 437: Reference to Figure 4 is missing.
We are sorry for the missing reference and very much appreciate the attentiveness of the reviewer. We included the missing reference
Line 513: The figure reference needs to be corrected to Figures 6a,b.
We are sorry for this mistake and very much appreciate the attentiveness of the reviewer. The mislabeling was corrected.
Please maintain consistency with the time units, sticking to either "days post immunization/infection" or "weeks" throughout the manuscript (e.g., line 388-391 and line 429).
This was a good suggestion and we corrected the time units to finally maintain consistency with “days” as consistent term. In addition, we used the term “ days post vaccination (d.p.v.)”.
Overall, addressing these points and minor editing errors will significantly enhance the clarity and quality of the manuscript.
We greatly appreciate the reviewer’s positive response and for pointing out the need to further improve and clarify the discussion. We tried to corroborate our findings by citing these papers within the discussion section of the revised manuscript.

Round 2
Reviewer 1 Report
The authors have responded to the comments point-by-point. My main concern with the experimental design was limited assessment of immunogenicity prior to challenge and sampling not being conducted at the same time post-challenge. Instead, comparison between viral loads and pathology between controls and vaccinated (survivors) mice were done at time of euthanasia due to severe clinical signs (for survivors at the end of the study period). The authors are now acknowledging this by adding statements to the manuscript. Also stating that follow-up experiments would be required to better assess this. I would be in favor of approving the manuscript for publication with these changes.
Author Response
Reviewer Comments:
Reviewer 1:
The authors have responded to the comments point-by-point. My main concern with the experimental design was limited assessment of immunogenicity prior to challenge and sampling not being conducted at the same time post-challenge. Instead, comparison between viral loads and pathology between controls and vaccinated (survivors) mice were done at time of euthanasia due to severe clinical signs (for survivors at the end of the study period). The authors are now acknowledging this by adding statements to the manuscript. Also stating that follow-up experiments would be required to better assess this. I would be in favor of approving the manuscript for publication with these changes.
We greatly appreciate the reviewer’s overall positive and very constructive response which enabled us to submit a clearly improved manuscript. To consider the reviewers’ concern, we included the following sentence within the conclusion: “In this context, follow-up experiments will be required to analyze the immunogenicity of the different MVA-NP candidate vaccines prior to challenge in more detail. Such future studies should also analyze viral load in the organs at the same time post-challenge infection.”
